# Microspheres Used in Liver Radioembolization: From Conception to Clinical Effects

**DOI:** 10.3390/molecules26133966

**Published:** 2021-06-29

**Authors:** Philippe d’Abadie, Michel Hesse, Amandine Louppe, Renaud Lhommel, Stephan Walrand, Francois Jamar

**Affiliations:** Department of Nuclear Medicine, Cliniques Universitaires Saint Luc, Université Catholique de Louvain, 1200 Brussels, Belgium; michel.hesse@uclouvain.be (M.H.); amandine.louppe@uclouvain.be (A.L.); renaud.lhommel@uclouvain.be (R.L.); stephan.walrand@uclouvain.be (S.W.); francois.jamar@culouvain.be (F.J.)

**Keywords:** liver radioembolization, radiolabeled microspheres, dosimetry

## Abstract

Inert microspheres, labeled with several radionuclides, have been developed during the last two decades for the intra-arterial treatment of liver tumors, generally called Selective Intrahepatic radiotherapy (SIRT). The aim is to embolize microspheres into the hepatic capillaries, accessible through the hepatic artery, to deliver high levels of local radiation to primary (such as hepatocarcinoma, HCC) or secondary (metastases from several primary cancers, e.g., colorectal, melanoma, neuro-endocrine tumors) liver tumors. Several types of microspheres were designed as medical devices, using different vehicles (glass, resin, poly-lactic acid) and labeled with different radionuclides, ^90^Y and ^166^Ho. The relationship between the microspheres’ properties and the internal dosimetry parameters have been well studied over the last decade. This includes data derived from the clinics, but also computational data with various millimetric dosimetry and radiobiology models. The main purpose of this paper is to define the characteristics of these radiolabeled microspheres and explain their association with the microsphere distribution in the tissues and with the clinical efficacy and toxicity. This review focuses on avenues to follow in the future to optimize such particle therapy and benefit to patients.

## 1. Introduction

Liver radioembolization (RE) is commonly used for the treatment of hepatocellular carcinoma and secondary liver malignancies. This treatment is performed by injection of radioactive microspheres in the liver artery after transarterial catheterization. Radioactive microspheres are trapped in the microvasculature (arterioles) of tumors and of the liver parenchyma. Unlike the liver parenchyma, where blood supply is almost obtained by the portal vein, liver tumors are preferentially vascularized by the liver arteries. This preferential perfusion allows us to achieve good targeting of hypervascularized tumors with a limited radiation of the non-tumoral liver [1]. The technique was initially developed using iodine-131 (^131^I) Lipiodol, a radiolabeled ethiodized oil [2]. Thereafter, radiolabeled microspheres have emerged using Yttrium-90 (^90^Y) and holmium-166 (^166^Ho). These radionuclides emit beta radiations of high energy, resulting in a high delivery of energy to the tumors (absorbed doses) in the range of 100 to 1000 Gy [3]. In comparison, the total tumor dose is limited to a maximum of 70 Gy in external beam radiotherapy to avoid liver damage [4].

Three types of microspheres are commercially available: ^90^Y-resin microspheres (Sir-Spheres^®^, Sirtex Medical Ltd., Sydney, Australia), ^90^Y-glass microspheres (Therasphere^®^, Boston Scientific, Boston, MA, USA) and ^166^Ho-poly-L-lactic acid (PLLA) microspheres (QuiremSpheres^®^, Quirem Medical B.V., Deventer, The Netherlands). Other microspheres have been used but did not reach the level of marketing authorization in Europe, e.g., ^188^Re-microspheres [5].

A simulation of treatment always precedes RE for evaluating the feasibility of the treatment [6]. This first procedure includes an arteriography with an accurate mapping of the liver arterial anatomy and the arterial branches of the tumor(s). Coil embolization(s) of some arterial branches is sometimes performed to avoid the risk of microsphere deposition in the digestive tract during the treatment phase. Finally, a diagnostic radiopharmaceutical agent, technetium-99m macro aggregated albumin (MAA), is injected in the arterial territory of the targeted tumors, simulating the injection of radioactive microspheres. Nuclear imaging with abdominal SPECT/CT follows this arteriography for evaluating the MAA distribution. MAA imaging permits us to confirm the good targeting of the tumor(s) and rule out extrahepatic abdominal uptake and lung shunting. In some centers, the MAA distribution in the tumor and the normal liver compartments is also analyzed precisely to perform a more personalized calculation of the activity needed for treatment (partition model) [7]. To the contrary of resin and glass microspheres, PLLA microspheres can also be used as scout dose during this workup phase. This scout dose (maximum 250 MBq ^166^Ho) replaces the tasks of MAA SPECT/CT without significant toxicity [8].

^90^Y-resin microspheres, ^90^Y-glass microspheres and ^166^Ho-PLLA microspheres have different physico-chemical characteristics and different biological effects. This review evaluates how these characteristics impact the microsphere distribution in the tissues and to what extent it may explain the clinical effects.

## 2. Preparations and Labeling (Sir-Spheres^®^, Therasphere^®^, QuiremSpheres^®^)

SIR-Spheres^®^ are cation exchange resin microspheres labeled with ^90^Y phosphate. The resin is supplied as symmetrical microspheres ranging from 30 to 50 µm in diameter, made of sulphuric acid groups attached to a styrene divinylbenzene copolymer resin (Aminex 50W-X4, Bio-Rad, Hercules, CA, USA). First, stable yttrium oxide (III) (^89^Y) is activated in a neutron beam to its radioisotope ^90^Y according to the nuclear reaction ^89^Y (n,γ) ^90^Y. The radioactive yttria (Y_2_O_3_) is then dissolved in sulphuric acid. ^90^Y is adsorbed onto the resin matrix by adding newly formed ^90^Y sulphate solution to the aqueous slurry of microspheres. To immobilize and stably incorporate the ^90^Y into the lattice, the radionuclide is precipitated as an insoluble phosphate salt by adding a tri-sodium phosphate solution. Finally, the microspheres are washed with a phosphate buffer solution and resuspended with water for injection [9].

The final product contains approximately 45 million microspheres per vial with a maximum specific activity of 75 Bq per microsphere [10].

SIR-Spheres^®^ is supplied as a 3 GBq ± 10% dose, calibrated on the planned treatment date, and can be used up to 24 h after calibration. The patient dose is dispensed on-site in the nuclear pharmacy [11].

The second commercially available product labeled with ^90^Y is TheraSphere^®^. It consists of radioactive yttrium oxide-aluminosilicate glass microspheres. Inactive yttria is mixed with aluminum and silicone oxide. The resulting mixture is melted and stirred in a furnace at 1500 °C. The melt is then removed from the heat and rapidly quenched to a glass. The resultant glass frit is crushed to fine powder and filtered through a sieve. The powder is spheroidized by introducing it into a flame: the particles are melted and formed into spherical liquid drops by surface tension. The newly formed microspheres are quickly cooled and screened. The 20- to 30-µm fraction is rinsed and dried. The ^89^Y embedded glass microspheres are eventually activated by neutron bombardment to the radioisotope ^90^Y [10,12,13].

The final product has a specific activity of 2500 Bq per sphere at the time of calibration, a density of 3.3 g/mL and a diameter ranging from 20- to 30-µm.

TheraSphere^®^ is supplied in six activity sizes—3-, 5-, 7-, 10-, 15-, and 20-GBq—with possible customized sizes [13]. The activity ordered is based on the ^90^Y decay, the required patient specific activity and the planned treatment date. It is delivered in a sealed vial, without preparation on-site. Therefore, TheraSphere^®^ must be administered at a specific time.

QuiremSpheres^®^ are poly-L-lactic acid (PLLA) microspheres labeled with ^166^Ho. The microspheres are synthesized by the solvent-evaporation technique. First, non-radioactive ^165^Ho is complexed with acetylacetonate. Complex ^165^Ho-acetylacetonate is then incorporated into the PLLA matrix by adding ^165^Ho-actylacetonate and poly-L-lactic acid to a continuously stirred chloroform solution. The resulting solution is then added to an aqueous polyvinyl alcohol solution and stirred until complete evaporation of the chloroform. The newly formed ^165^Ho-loaded microspheres are collected by centrifugation, washed and fractionated according to their size. The 20- to 50-μm fraction is dried and packed in polyethylene vials. Finally, ^165^Ho is activated to ^166^Ho in a nuclear reactor by neutron irradiation with a thermal neutron flux of 5 × 10^12^ cm^−2^ s^−1^ (^165^Ho + n→^166^Ho, cross section 64 barn) [14,15,16]. The final product contains 19% (*w*/*w*) holmium (essentially stable which is of interest in view of the paramagnetic properties of ^165^Ho, see paragraph 2) and a maximum ^166^Ho specific activity per microsphere of 450 Bq [17]. QuiremSpheres^®^ is supplied as a ready-to-use, tailored dose: the dispatched activity is patient-specific and matches the activity at treatment time as ordered by the customer. There is no need for patient-dose preparation on-site [18].

SIR-Spheres^®^, TheraSphere^®^ and QuiremSpheres^®^ received European regulatory approval (CE mark) as Active Implantable Medical Devices (Council Directive 90/385/EEC) in October 2002, June 2006 and April 2015 respectively.

Characteristics of the three commercially available microspheres are presented in Table 1 and Table 2.

## 3. Radionuclide Properties and Clinical Applications

Table 1 summarizes the main physical characteristics of ^90^Y and ^166^Ho radionuclides. Both emit beta particles of high energy in a similar mean range of 2.5 mm. More than 90% of the radiations are delivered during the first 11 days for ^90^Y (half-life: 2.7 days) and in only 4 days for ^166^Ho (half-life: 1.1 days) [19,20].

These beta particles generate free radical species in the presence of oxygen inducing DNA breaks and cell killing. Many factors influence the killing efficacy, and particularly the rate of beta particles emission (correlated to half-life) and the number of beta particles (correlated to the radioactive activity) [21]. The biological effect of radiations is demonstrated by the absorbed dose (Gy), defined by the energy (J) deposited per mass of tissue or tumor (kg). This absorbed dose is directly correlated to the cell survival fraction and tumor response [22].

The precise localization of these beta particles in the tumor is also an important factor for tumor response. Indeed, the very short range of beta emission (few millimeters) induce a biologic effect only in the neighborhood of the beta particles and consequently, their distribution in the tumor must be the as homogeneous as possible for killing a maximum of cells and induce a tumor response [23].

These radionuclides have also the ability to be detected by nuclear imaging systems. An accurate detection is important for evaluating the distribution in the targeted tissues and for a quantitative assessment of the dose deposition. Beta particles induce interactions with the surrounding tissues and generate a continuous spectrum of X-ray photons known as bremsstrahlung. These secondary photons can be captured by single-photon emission computed tomography with (X-ray) computed tomography (bremsstrahlung SPECT/CT) [24]. A more accurate imaging could also be performed by detection of the direct radiations emitted from ^90^Y and ^166^Ho (Table 1). ^90^Y emits 32 positrons for one million decays. This very low rate of positrons is, however, sufficient to obtain images of good quality and resolution using PET systems (^90^Y PET/CT) [25,26]. ^166^Ho emits a low abundance of primary gamma radiations (Table 1) detected also with acceptable accuracy by SPECT systems [20,27]. Holmium metal (as Table 1 ^65^Ho) has also the property to be highly paramagnetic and well detected by MRI systems [20]. Then, a quantitative assessment of the ^166^Ho biodistribution, indirectly by its stable carrier, could also be accurately provided with MRI [28].

## 4. Radioactive Microspheres Properties

^90^Y-resin, ^90^Y-glass and ^166^Ho-poly-L-lactic-acid [PLLA] microspheres differ from their physical characteristics, summarized in Table 2 [10,29]. Globally, all three devices have a similar diameter around 30 μm, but some differences appear in the size spectrum distribution (Figure 1). MAA particles have a significantly lower size (mean: 15 μm).

Resin and PLLA microspheres have a density comparable to blood (1.06 g/mL). Glass microspheres have a higher density (3.3 g/mL) three-fold more than blood. Regarding the specific activity per microsphere and the number of microspheres injected, resin, glass and PLLA microspheres differ highly (Table 2).

The specific activity per sphere is respectively 50 and 5 higher for glass microspheres as compared to resin and PLLA microspheres. Moreover, the maximal activity per vial differs from 20 GBq for glass, 15.1 GBq for ^166^Ho-PLLA and 3.2 GBq for resin microspheres (one week after calibration) [33,34]. The number of microspheres also differs between formulations: approximately 10 million for PLLA, 40 million for resin and only 4 million for glass microspheres (for a 3 GBq activity at the time of treatment). Methods to calculate the activity delivered for the treatment also differ and are summarized in Table 3. The calculation methods for activity planning are semi empirical and were developed by the manufacturers to limit the toxicity to the healthy liver (diffuse liver damage). For resin microspheres, the most commonly used method is based on the body surface area (BSA), assuming a linear correlation between the liver size and the BSA. For glass and PLLA microspheres, the activity planning is based on the Medical Internal Radiation Dose principles and aims to reach a non-toxic dose to the liver [35]. Compared to ^90^Y-resin microspheres, the administered activity is usually threefold higher for ^166^Ho-PLLA and 2.5-fold higher for glass microspheres [34,36].

## 5. Impact at a Microscopic Level-Microdosimetry

Due to their physical differences, these types of microspheres have a different distribution into the liver at the microscopic level, which results in major differences in local absorbed doses.

Investigations of the microscopic depositions of microspheres can be investigated in vivo by imaging (SPECT, PET and MRI) or simulated in vitro by computational or artificial models of the hepatic arterial system. An excellent agreement was recently demonstrated between the microscopic deposition of microspheres simulated with a computational model and the real distribution observed in patients with ^90^Y PET/CT [37], especially with time- of-flight (TOF)-PET systems [38,39].

Contrarily to external radiotherapy, the absorbed dose distribution is heterogeneous in liver radioembolization resulting from the heterogeneous microsphere distribution at a microscopic level [40,41]. Moreover, due to their physical differences, the distribution of the absorbed dose into the liver is very different between the types of microspheres.

^90^Y-resin, ^90^Y-glass and ^166^Ho-PLLA microspheres have a similar diameter permitting us to similarly reach the microvasculature of tumors. The higher density of glass microspheres (Table 2) does not have a significant impact in the microsphere distribution compared to resin microspheres, as demonstrated by an experimental model of the hepatic vasculature [42]. However, Jernigan et al. investigated the effects of density and demonstrated a lower penetration of glass compared to resin microspheres in a surrogate hepatic arterial system [43]. In their flow model, the distal penetration depth was significantly higher for resin microspheres at similar injection rates. This difference could be theoretically responsible of an reduced coverage of the tumor.

The main differences between microspheres are the specific activity per sphere and the number of injected microspheres during a treatment. Glass microspheres are especially highly radioactive (2500 Bq) and in limited number compared to others. Using a computational model, Walrand et al. demonstrated an asymmetrical distribution of the microspheres in the arterial liver tree, resulting in a large nonuniformity in the microsphere deposition at a microscopic level, especially when the number of microspheres is limited [38]. In a very interesting in-vivo study in pigs, Pasciak et al. confirmed in-vivo correlation between the number of injected microspheres and the degree of homogeneity of the absorbed dose at a microscopic level [39]. In this study, four pigs received lobar infusions of ^90^Y glass microspheres with an activity reaching a similar target liver dose of 50 Gy. In each pig, the injected glass microspheres differ in specific activity and number (from 1532 Bq/microsphere & 610 microspheres/mL of tissue to 70 Bq/microsphere & 15,555 microspheres/mL of tissue), as illustrated in Figure 2.

The analysis of the microdosimetry data demonstrated important differences in absorbed doses deposition at a microscopic level. Each pig received the same average dose of 50 Gy, but the distribution at a microscopic level was clearly different, being very inhomogeneous when the number of microspheres was lower. The percentage of the non-target liver receiving at least a potentially toxic dose of 40 Gy was only 24% for microspheres in relatively small number (e.g., 610 microspheres/mL of tissue) and 53% using more numerous microspheres (e.g., 15,555 microspheres/mL of tissue). This effect is well illustrated in Figure 3.

Using microspheres in limited number induces a more heterogeneous liver distribution, but also permits it to be less toxic, avoiding reaching a toxic dose in a large part of the normal liver. These dosimetric concepts could be extrapolated to any type of microspheres, in order to evaluate to what extent the number of microspheres and their specific activity can influence the efficacy and toxicity of liver radioembolization.

Using a Monte Carlo modeling, Pasciak et al. also demonstrated a similar behavior in tumors [44]. The microsphere density was directly correlated with the tumor absorbed dose homogeneity. In this model, no significant differences were observed for microsphere density in the range of 10,000–70,000 spheres/mL of tissue, but apparent differences appear below 5000 spheres/mL of tissue, resulting in more heterogeneous tumor dose distribution and risk of ineffectiveness. In this study, they also simulated the minimal dose deposited in 70% of the tumor (D70) and demonstrated a decrease up to 20% for the lowest number density (200 spheres/mL of tissue). Very interestingly, they also estimated the microsphere number density in tumors receiving an absorbed dose of 100 Gy or 250 Gy with ^90^Y-resin microspheres and ^90^Y glass microspheres. For a tumor absorbed dose of 100 Gy with ^90^Y resin microspheres, the microsphere number density was largely above the critical number density of 5000 spheres/mL but came under the limit for ^90^Y glass microspheres with a range between 820 and 18,600/mL. However, using a tumor-absorbed dose of 250 Gy, the microsphere number density became optimum in a range between 2050 and 46,500/mL for ^90^Y glass microspheres. Therefore, by targeting a higher tumor absorbed dose (>100 Gy) with ^90^Y glass microspheres, the dose distribution at a microscopic level is more homogeneous, reaching an efficient dose deposition in a large part of the tumor.

^90^Y- resin and^166^Ho- PLLA microspheres could have an embolic effect because a high number of particles are injected in the liver arterial tree (Table 2). However, this effect may be very limited in particular because they have a very small size, permitting to reach the terminal microvasculature [45]. Moreover, Bilbao et al. demonstrated no significant ischemic effect in pig liver models after injection of large quantities of non-radioactive resin microspheres [46]. This is also explained by the unique dual blood supply of the normal liver, provided mainly by the portal vein.

## 6. Impact at a Macroscopic Level-Clinical Effects

Physical characteristics of the radioactive microspheres explain also their different toxicity and efficacy profiles. Regarding PLLA microspheres (Quirem Spheres^®^), more data are needed to evaluate and compare its efficiency and toxicity.

Regarding toxicity, an excessive irradiation of the healthy liver can induce a severe and potentially life-threatening complication: radioembolization induced liver disease (REILD) with an incidence rate inferior to 4% [45]. It is described by a sinusoidal obstruction syndrome and by a liver damage within 3 months after RE, in absence of tumor progression [47]. This complication tends to occur especially when a large volume of liver parenchyma is exposed to radiations. Other risk factors include a recent exposition to chemotherapy and underlying cirrhosis [48]. In external beam radiotherapy (EBRT), this complication occurs with whole liver doses of 30–35 Gy [49]. The tolerance is higher in RE, 40–50 Gy for resin microspheres and 90 Gy with glass microspheres [50,51,52]. A pre-clinical study in pigs with administration of PLLA microspheres demonstrates no toxicity with absorbed doses over 100 Gy. In human studies in phase 1 and 2, the whole liver absorbed dose was limited to 60 Gy and was not associated with any liver toxicity [53,54]. The homogeneity of the dose distribution explains the relative lower liver dose tolerability of external radiotherapy. To the contrary, a large heterogeneity of the dose distribution appears in RE, and a relative less proportion of liver parenchyma receives a toxic dose (i.e., 30–35 Gy). As described by Pasciak et al. [39], this liver dose heterogeneity is directly correlated to the number of microspheres injected in the liver. This characteristic could explain why the tolerable dose is significantly higher when a relatively small number of radioactive microspheres are injected into the liver (Table 2).

The liver is a pure parallel organ made of independent (≈1.5 mm-size) subunits called lobules. Similarly to tumor, the liver does not contain stem cells, the liver cells having the capacity to indefinitely divide while carrying out their metabolic task. Centimetric scale heterogeneity pattern in the activity distribution post radioembolization were evidenced by independent modalities: ex-vivo microscopy analysis of liver tissue resection [41], ^166^Ho loaded spheres MRI imaging [55] and Monte Carlo simulations of spheres transport in the hepatic arterial tree [38,56].

The ^90^Y beta range being sub-centimetric, the result is that, contrary to EBRT, the absorbed dose itself is highly heterogeneous. As a consequence, for an absorbed dose of 40 Gy, a sufficient fraction of lobules experienced sub-lethal dose in TARE and can repopulate the liver pool. Estimation of this fraction by Monte Carlo simulations explains the difference between the maximal safe absorbed dose between glass and resin spheres radioembolization, as well as EBRT [57].

The same effect holds for the tumors. Jones and Hoban [58] developed the equivalent uniform dose (EUD) formalism to take into account this heterogeneity effect in tumor. The EUD computed from post ^90^Y radioembolization TOF-PET imaging reunifies the resin and glass microsphere patient survival observed post glass and resin spheres radioembolization [59].

Another possible complication is the development of gastric/ duodenal ulcers, reported in 3.1% with resin microspheres and 0.1% with glass microspheres [60]. A gastrointestinal deposition of microspheres leads to an ulceration induced by radiations. It can occur when the liver arteries communicate with the digestive tract by collaterals or when the microsphere injection is administered in close digestive arteries such as the gastro duodenal artery or the right gastric artery [61]. Prophylactic coil embolization of the hepato-splanchnic arteries or more distal injections of microspheres can be performed to prevent this complication [45]. The main risk factor for gastro-duodenal ulceration is a stasis of the blood flow during the microsphere administration causing retrograde flux and extra-hepatic deposition [62]. Due to the high number of injected microspheres (Table 2), a potential stasis can occur during the administration of resin or PLLA microspheres [63]. With resin microspheres, an early stasis may occur in 20% of the procedures, resulting in an incomplete delivery of the prescribed activity and a suboptimal dose to tumors [64]. Conversely, no evidence of flow stasis was demonstrated with glass microspheres, especially because a relatively small number of microspheres is injected (Table 2) [65]. These physical properties also have a significant impact in the treatment approach. A solitary tumor isolated to one or two contiguous liver segments could be treated by a radical approach, performing a radiation segmentectomy. This approach is performed by delivering a high activity of radioactive microspheres in a small volume. The ablative dose to the targeted liver could be easily reached using glass microspheres but could not be well achieved using resin or PLLA microspheres due to the flow stasis during the administration. To solve this limitation, the Sirtex FLEXdose delivery program was recently purposed permitting to order resin microspheres with physical properties closer to glass microspheres (lower number of microspheres of higher specific activity).

RE is being part of the treatment strategy of unresectable hepatocellular carcinoma (HCC) and liver metastases, especially from colorectal and neuroendocrine origin [66]. For HCC, comparative studies did not demonstrate differences in response rates and patient outcome between resin and glass microspheres [67,68]. No data are currently available to evaluate the efficacy of PLLA microspheres in HCC. A disease control rate of 90% at 6 months was reported in a little retrospective study enrolling 9 patients [69]. Similar survival rates were also shown using resin or glass microspheres in colorectal liver chemo-resistant metastases [70,71]. A phase 2 study with 23 patients treated with holmium-166 microspheres in salvage therapy of colorectal liver metastases also demonstrated similar outcome (overall survival: 13.4 mo versus 8.3–15.2 mo with ^90^Y-microspheres) [53,72].

Previous data demonstrated a continuous relationship, a sigmoid correlation, between the tumor absorbed dose and the radiological response [51,73]. HCC patients treated with resin microspheres had better disease control and overall survival when tumor absorbed doses were above 100 Gy. HCC patients treated with glass microspheres also had a better outcome with a tumor-absorbed dose of at least 205 Gy [74,75]. For colorectal metastases, a dose response relationship was also demonstrated with the three types of microspheres. A significant metabolic response was achieved with a tumor cutoff dose of 50–60 Gy with resin microspheres [76,77], 139 Gy using glass microspheres [78] and 90 Gy using PLLA microspheres [79]. Then, to be efficient, the tumor absorbed dose cutoff is approximately twice with glass microspheres as compared to resin microspheres. A recent comparison of patients treated with glass and resin microspheres demonstrated similar outcome using a tumor dose cutoff of 61 Gy with resin microspheres and 118 Gy with glass microspheres [80]. But interestingly, the analysis of the tumor dose at a microscopic level did not demonstrate a two-fold difference between both types of microspheres. Indeed, patients treated with glass or resin microspheres achieved similar outcome with the same minimal dose in most of the tumor volume. A minimum dose of 40 Gy in 66% of the tumor volume was associated with similar PFS and OS with both types of microspheres. The tumor dose distribution can be accurately studied with ^90^Y PET/CT using dose volume histogram (DVH). Figure 4 demonstrates examples of HCC tumors treated with glass microspheres with similar mean absorbed doses, but wide differences in intratumoral dose distributions.

Similarly, tumor absorbed doses could be corrected for heterogeneity distribution using Equivalent Uniform Doses (EUD). Similar patient outcome was demonstrated with glass and resin microspheres with the same EUD cutoff of 40 Gy [59]. The higher efficacy cutoff dose of glass microspheres is explained by the more heterogeneous distribution of the microspheres. As previously described in the dosimetric model of Pasciak et al. [44], a similar tumor dose distribution could be achieved using a greater number of microspheres with a lower specific activity (i.e., ^90^Y-resin microspheres) or using fewer microspheres of greater specific activity (i.e., ^90^Y-glass microspheres) and by increasing the mean absorbed dose (by increasing the administered activity).

## 7. Future Directions

Knowledge in RE has been considerably improving over the last decade. The better understanding of the microdistribution of the microspheres and of their radiobiologic effects permits to notably optimize the technique. The dose-response and dose-toxicity relationships have been clearly and accurately demonstrated. Standard methods for activity planning (Table 3) are progressively replaced by more personalized methods that consider the dose deposition in the tumor and non-tumoral compartments in order to improve the clinical benefits of RE. The DOSISPHERE study (performed with glass microspheres) recently demonstrated significant improvement in response rates using a personalized dosimetry with a targeted tumor dose of at least 205 Gy in HCC [75]. Recent recommendations with resin microsphere also consider performing a personalized dosimetry targeting a tumor dose of at least 100–120 Gy [81]. Moreover, the toxicity cutoffs doses are well determined, and the treatment plan can be optimized in order to deliver the maximal tolerable absorbed dose to the non-tumoral liver [52]. With PLLA microspheres, the development of the ^166^Ho scout dose may significantly optimize the accuracy of the treatment planning [82]. MAA particles have a significantly smaller size than PLLA microspheres which may induce an overestimation of the lung shunt with MAA (Figure 1). The predictive tumor dosimetry was more accurate using the ^166^Ho scout dose compared to the MAA-based dosimetry [83]. This holds true though if every effort is dedicated to repositioning the catheter exactly in the same place during the therapeutic arteriography. Moreover, innovations in interventional radiology permit also to better target the liver tumors. C-arm cone-beam computed tomography (CBCT) is more and more used to analyze precisely the vascular territories of the tumor(s) [84]. Micro catheters also permit performing liver arteriographies at a segmental level to deliver very high doses to tumors and preserving the non-tumoral parenchyma [85]. Furthermore, a recent analysis of the microsphere deposition using innovative antireflux catheters demonstrated a significant improvement of the tumor deposition of microspheres as compared to classic end-hole catheters [86]. Moreover, infusion of a mesenteric arterial vasodilator such as dopexamine during the procedure of RE could widely improve, in theory, the tumor targeting, and hence the treatment benefits [87]. Other in-situ radiotherapeutic approaches—for instance, the recently developed phosphorus-32 microparticles (Oncosil^TM^ system, Sydney, Australia) for treating biliary and pancreatic tumors—are injected directly in the tumor(s), with imaging guiding. Preliminary results are encouraging for locally advanced pancreatic cancer [88].

## Figures and Tables

**Figure 1 molecules-26-03966-f001:**
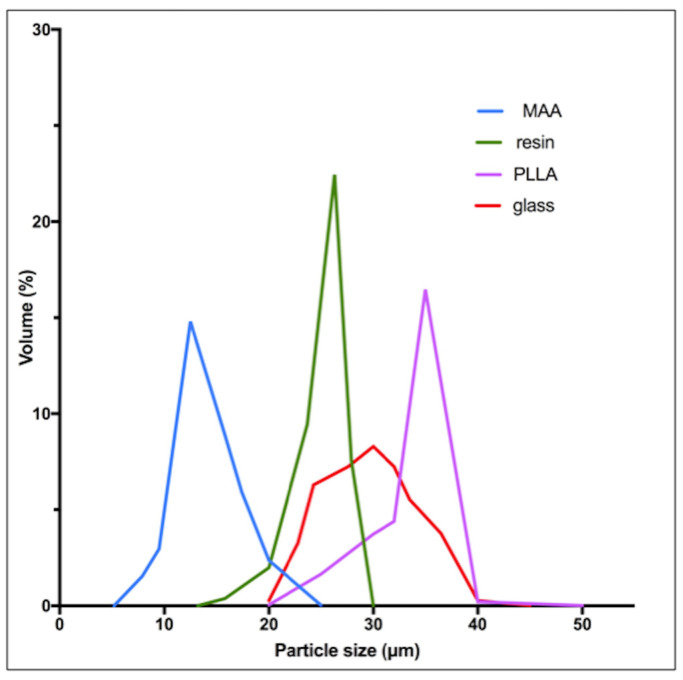
Particle size distribution for resin, glass, PLLA microspheres and MAA particles. Figure 1 was made by rescaling graphs derived from the analyses of Bakker et al. [30], Bult W. [31] and Gupta et al. using iron labeled glass microspheres [32].

**Figure 2 molecules-26-03966-f002:**
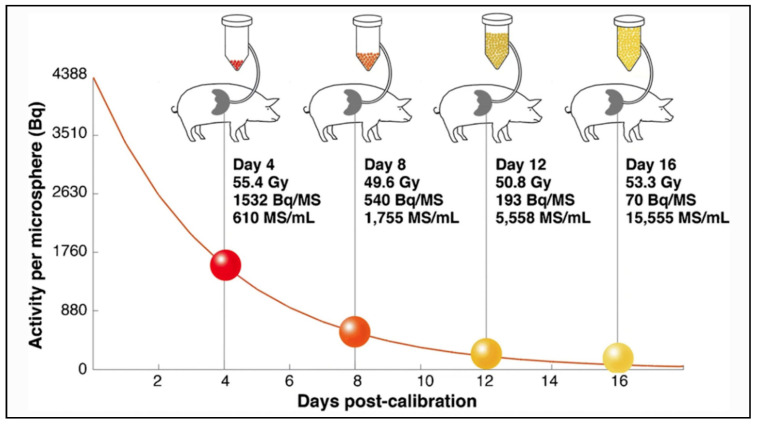
Amount of glass microspheres injected in pig livers as a function of time from microsphere calibration (4, 8, 12 or 16 days), resulting in spheres of different specific activity and concentrations. Reprinted from European journal of nuclear medicine and molecular imaging with permission of Springer Nature (*Eur. J. Nucl. Med. Mol. Imaging* 2020, 47, 816–827).

**Figure 3 molecules-26-03966-f003:**
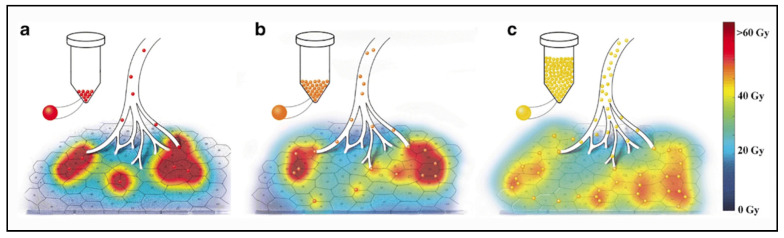
Artistic rendering of microscopic microsphere deposition at day 4 post-calibration (**a**), 8 post-calibration (**b**) and 12 post-calibration (**c**). Increased homogeneity of absorbed dose is apparent for glass microspheres injected at day 12 post-calibration (e.g., 193 Bq/microsphere, 5558 microspheres/mL). The proportion of the liver receiving a dose higher or equal to 40 Gy (red and yellow colors) was higher for glass microspheres injected at day 12 post-calibration (**c**). Reprinted from European journal of nuclear medicine and molecular imaging with permission of Springer Nature (*Eur. J. Nucl. Med. Mol. Imaging* 2020, 47, 816–827).

**Figure 4 molecules-26-03966-f004:**
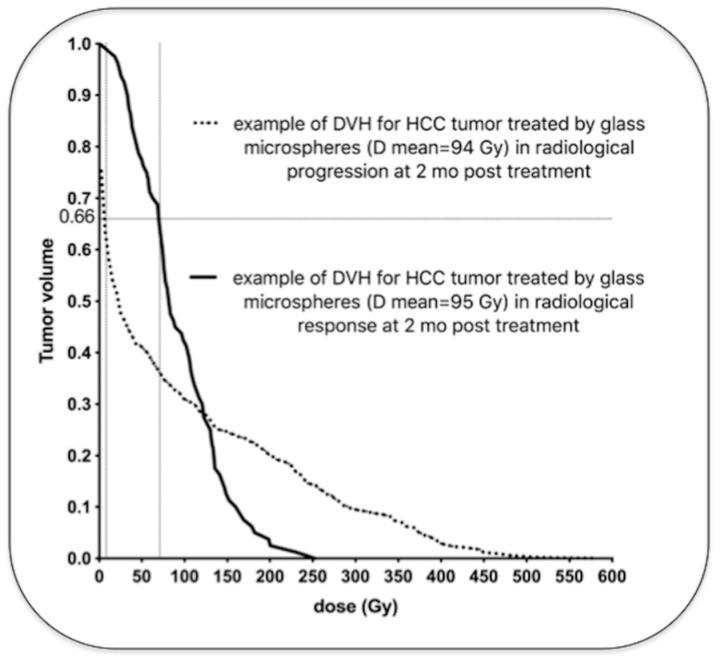
Examples of two HCC tumors treated with glass microspheres with similar mean absorbed doses (94 and 95 Gy) but very different dose distributions (personal data). One tumor (dotted line) received a very low dose in a large part of the tumor volume (more than 5 Gy in 66% of the tumor volume), and was in radiological progression two months later. The other tumor (solid line) received a high dose in a large part of the tumor volume (more than 69 Gy in 66% of the tumor volume) and was in radiological response two months later. Note that the cluster with the highest local dose was found in the non-responding tumor (>500 Gy) because of the major heterogeneous distribution.

**Table 1 molecules-26-03966-t001:** Physical characteristics of radionuclides.

	Labeled Microspheres	Half-Life	Beta Emission (E. Max)	Range of Beta Radiation	Other Emissions
Yttrium-90 (^90^Y)	Resin or glass	2.7 days	2.28 MeV	Mean: 2.5 mm (Max: 11 mm)	Positron (32 × 10^−6^)
Holmium-166 (^166^Ho)	Poly (L-Lactic acid)	1.1 day	1.77 MeV (49%), 1.86 MeV (50%)	Mean: 2.5 mm (Max: 8.7 mm)	Gamma (6.7%)

**Table 2 molecules-26-03966-t002:** Physical characteristics of radioactive microspheres.

Microspheres	Diameter (Mean)	Density	Approximative Number of Micro-Spheres Per GBq *	Activity Per Microsphere
^90^Y-Resin	32 μm	1.6 g/mL	13 × 10^6^	50 Bq
^90^Y-Glass	25 μm	3.3 g/mL	0.4 × 10^6^	2500 Bq
^166^Ho-PLLA	30 μm	1.4 g/mL	10 × 10^6^	450 Bq

* On the day of calibration (approximately). PLLA: poly-(L-lactic-acid).

**Table 3 molecules-26-03966-t003:** Activity planning.

Types	Usual Recommended Method for Activity Planning
^90^Y-Resin (BSA method)	$A\left(\mathrm{GBq}\right)=\left(\mathrm{BSA}-0.2\right)+\frac{{\mathrm{Volume}}_{\mathrm{tumor}}}{{\mathrm{Volume}}_{\mathrm{tumor}}+{\mathrm{Volume}}_{\mathrm{normal}\;\mathrm{liver}}}$ (BSA: the body surface area in m^2^)
^90^Y-Glass (mono-compartmental method)	$A\;\left(\mathrm{GBq}\right)=\frac{{\mathrm{Dose}}_{\mathrm{liver}}\cdot {\mathrm{Mass}}_{\mathrm{liver}}}{50}$ (Dose_target liver_: 80–150 Gy)
^166^Ho-PLLA (mono-compartmental method)	$A\left(\mathrm{GBq}\right)=\frac{\;{\mathrm{Dose}}_{\mathrm{whole}\;\mathrm{liver}}\cdot {\mathrm{Mass}}_{\mathrm{whole}\;\mathrm{liver}}}{15.9}$ (Dose_whole liver_: 60 Gy)

## Data Availability

Data sharing not applicable. No new data were created or analysed in this study.

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
