# Peer review of "Microspheres Used in Liver Radioembolization: From Conception to Clinical Effects"

_molecules, 2021, doi:10.3390/molecules26133966_

Round 1

Reviewer 1 Report

Concerning with my overall recommendation upon this review, the subject addressed is quite worthy of investigation and dissemination. The scientific information presented is relativelly new, innovative and relevant for dissemination in scientific literature. The organization of the manuscript is appropriate and detailed in order to reach a better understanding of the physicochemical properties, microdistribution and of radiobiologic effects of the radioactive microspheres in order to permits to get more understanding and to contribute optimizing the technique.

Further, tables, figures and the level of the English of the manuscript are appropriate. Finally, the conclusions are supported by the data rationalized from the most relevant references of the state of the art, and therefore the manuscript being  appropriated for the scope of the journal.

Therefore, I recommend this work to be published in Molecules.

Author Response

We thank you very much for reviewing this article.

Reviewer 2 Report

In this article, Philippe D'ABADIE et al have present an relatively comprehensive and well written review of three commercially available microspheres for liver radioembolization. After minor revision, this article should be published. 

Here are some suggestions:

  1. The preparations and design of the microspheres largely determines their outcoming physical and chemical properties. It would particularly helpful to have graphic demonstrations for the design of each individual microspheres.
  2. It would be helpful to name and compare the application scenarios and limitations for each microspheres.
  3. In section 3, please clarify the reason why to choose different calculation methods for activity planning. 
  4. In section 4, the discussion focused on 90Y-glass, while more detailed discussion about 90Y-resin and 166Ho-PLLA will be appreciated. 

In addition, some minor issues: 

  1. Extra spaces were found in between sentences. E.g. line 231, 251, 258 270, 319 and etc.
  2. Title of section 5, extra "L" for "Level".
  3. Line 292 and 293, please use "tumor" as unified spelling for this article instead of "tumour".
  4. Line 335, use "two-fold".
  5. Line 349, missing a period before "Note".

Author Response

Dear reviewer,

We thank you very much for reviewing this manuscript. This is a point-by-point reply to each comment.

In this article, Philippe D'ABADIE et al have present an relatively comprehensive and well written review of three commercially available microspheres for liver radioembolization. After minor revision, this article should be published. 

  • 1- The preparations and design of the microspheres largely determines their outcoming physical and chemical properties. It would particularly helpful to have graphic demonstrations for the design of each individual microspheres.

Author’s reply :

A new graph is added to the manuscript (section 3), demonstrating the size distribution of each type of microspheres. This following text is also added in the manuscript : « Resin, glass and PLLA microspheres have a similar size around 30 um but some differences appear in the size spectrum distribution (fig. 1). MAA particles have a significantly smaller size (mean: 15 μm). » In the last section, this sentence was also added : « MAA particles have a significantly smaller size than PLLA microspheres which may induce an overestimation of the lung shunt with MAA (figure 1). »

  • 2- It would be helpful to name and compare the application scenarios and limitations for each microspheres.

Author’s reply :

Despite the physico-chemical differences of each type of microspheres, clinical studies did not actually demonstrate any advantage for one type of microspheres. However, the physical properties of the different types of microspheres could result in some advantages or limitations depending on the liver volume targeted by radioembolization.

This text is added in the manuscript in section 5: « These physical properties have also a significant impact in the treatment approach. Solitary tumor isolated to one or two contiguous liver segments could be treated by a radical approach, performing a radiation segmentectomy. This approach is performed by delivering a high activity of radioactive microspheres in a small volume. The ablative dose to the targeted liver could be easily reached using glass microspheres but could not be well achieved using resin or PLLA microspheres due to the flow stasis during the administration. To solve this limitation, the Sirtex FLEXdose delivery program was recently purposed permitting to order resin microspheres with physical properties closer to glass microspheres (lower number of microspheres of higher specific activity). »

  • 3- In section 3, please clarify the reason why to choose different calculation methods for activity planning. 

     Author’s reply :

This text is added in section 3 :

« The calculation methods for activity planning are semi empirical and were developed by the manufacturers to limit the toxicity to the healthy liver (diffuse liver damage). For resin microspheres, the most commonly used method is based on the body surface area (BSA), assuming a linear correlation between the liver size and the BSA. For glass and PLLA microspheres, the activity planning is based on the Medical Internal Radiation Dose principles and aims to reach a non toxic dose to the irradiated liver. »

  • In section 4, the discussion focused on 90Y-glass, while more detailed discussion about 90Y-resin and 166Ho-PLLA will be appreciated. 

Author’s reply :

Section 4 focuses on microdosimetry and is mainly based on the original work of Pasciak et al. (Eur J Nucl Med Mol Imaging. 2020). In this work, they demonstrated the benefits and drawbacks performing treatments with different types of glass microspheres : a low number of microspheres of high specific activity versus an high number of micropsheres of low specific acticity. This work was exclusively performed with glass microspheres but the concepts developed in this paper could be extrapolated to other types of microspheres. A sentence was added in this section : «These dosimetric concepts could be extrapolated to any type of microspheres, in order to evaluate to what extent the number of microspheres and their specific activity can influence the efficacy and toxicity of liver radioembolization. »

  • In addition, some minor issues: 

1- Extra spaces were found in between sentences. E.g. line 231, 251, 258 270, 319 and etc. Author’s reply :done

  • Title of section 5, extra "L" for "Level".

Author’s reply :done

  • Line 292 and 293, please use "tumor" as unified spelling for this article instead of "tumour".

Author’s reply :done

  • Line 335, use "two-fold".

Author’s reply :done

  • Line 349, missing a period before "Note".
  • Author’s reply :done